# Pseudoprogression of Vestibular Schwannoma after Stereotactic Radiosurgery with Cyberknife^®^: Proposal for New Response Criteria

**DOI:** 10.3390/cancers15051496

**Published:** 2023-02-27

**Authors:** Daniel Rueß, Betina Schütz, Eren Celik, Christian Baues, Stephanie T. Jünger, Volker Neuschmelting, Alexandra Hellerbach, Markus Eichner, Martin Kocher, Maximilian I. Ruge

**Affiliations:** 1Department of Stereotaxy and Functional Neurosurgery, Center of Neurosurgery, Faculty of Medicine and University Hospital of Cologne, University of Cologne, Kerpener St. 62, 50937 Cologne, Germany; 2Department of Radiation Oncology, Cyberknife and Radiation Therapy, Faculty of Medicine and University Hospital of Cologne, University of Cologne, Kerpener St. 62, 50937 Cologne, Germany; 3Department of General Neurosurgery, Center of Neurosurgery, Faculty of Medicine and University Hospital of Cologne, University of Cologne, Kerpener St. 62, 50937 Cologne, Germany

**Keywords:** radiosurgery, vestibular schwannoma, Cyberknife^®^, skull base tumors, RANO criteria

## Abstract

**Simple Summary:**

After stereotactic radiosurgery of vestibular schwannomas, there may be a transient increase in tumor volume. Therefore, it is difficult to distinguish between tumor growth and treatment-related volume changes. To address this issue, we developed criteria to assess response by systematic volumetric analysis. We found an early (within the first 12 months after treatment) and a late (beyond 12 months) increase in volume. Consequently, in most cases with unclear volume increase after radiosurgery, longer observation intervals should be implemented to better distinguish between transient and continuous tumor growth.

**Abstract:**

(1) Background: Transient increase in volume of vestibular schwannomas (VS) after stereotactic radiosurgery (SRS) is common and complicates differentiation between treatment-related changes (pseudoprogression, PP) and tumor recurrence (progressive disease, PD). (2) Methods: Patients with unilateral VS (n = 63) underwent single fraction robotic-guided SRS. Volume changes were classified according to existing RANO criteria. A new response type, PP, with a >20% transient increase in volume was defined and divided into early (within the first 12 months) and late (>12 months) occurrence. (3) Results: The median age was 56 (range: 20–82) years, the median initial tumor volume was 1.5 (range: 0.1–8.6) cm^3^. The median radiological and clinical follow-up time was 66 (range: 24–103) months. Partial response was observed in 36% (n = 23), stable disease in 35% (n = 22) and PP in 29% (n = 18) of patients. The latter occurred early (16%, n = 10) or late (13%, n = 8). Using these criteria, no case of PD was observed. (4) Conclusion: Any volume increase after SRS for vs. assumed to be PD turned out to be early or late PP. Therefore, we propose modifying RANO criteria for SRS of VS, which may affect the management of vs. during follow-up in favor of further observation.

## 1. Introduction

Over the past decades, stereotactic radiosurgery (SRS) for small vestibular schwannoma (VS) has been established as an efficient and well-tolerated treatment, achieving high rates of tumor control and low risks for complications [1,2,3,4], and is therefore recommended in current guidelines [1,5]. However, in post-treatment MR images, a transient volume expansion (TVE) occasionally combined with central low signal intensity is frequently observed. So far, no clear criteria exist which define this phenomenon with respect to the percentage of volume increase or the observed time course. This lack of a clear definition can also be seen in the different terms used in the literature, e.g., “temporary enlargement” [6], “tumor expansion” [7,8], “transient expansion“ [9,10] or simply “pseudoprogression” [1,11,12]. Additionally, in many clinical situations it is unclear whether TVE reflects a treatment-related reaction or a true tumor recurrence. Particularly, in the case of Cyberknife^®^ SRS, only scarce data exist compared to the well-examined gamma-knife series. Thus, we prospectively analyzed the time course and extent of the tumor volume changes after Cyberknife^®^ SRS.

## 2. Methods

### 2.1. Study Population

Due to the retrospective nature of this investigation, approval from the ethical committee of the University of Cologne was waived (reference number 16-476). In this single center retrospective cohort study, we evaluated all patients undergoing single session robotic-guided SRS by Cyberknife^®^ for progressive vestibular schwannomas. Inclusion criteria were a minimum of at least two years of follow-up MR images after SRS. Clinical data were obtained through a review of the patients’ electronic medical record, the Cyberknife^®^ database, and available imaging studies. Data included patient demographic information, clinical history of neurological symptoms and parameters of the radiosurgical dose plan.

In general, follow-up imaging was scheduled at 6 and 12 months after SRS and annually thereafter. Earlier imaging was occasionally obtained based on clinical judgment and course.

### 2.2. Radiosurgery Technique

The radiosurgical technique for Cyberknife^®^-SRS has been described in detail in previous reports [2,3,13]. In brief, the tumor and the adjacent critical structures (e.g., brainstem, cerebellum, trigeminal nerve) were outlined by an experienced neurosurgeon on contrast enhanced, T1-weighted MR images (Phillips, MR-Scanner 1.5 or 3 Tesla), which were obtained prior to SRS and registered to a stereotactic planning CT (1 mm slice thickness, Toshiba 16-slice multidetector CT). The software Multiplan v4.5 was used for treatment planning. The final irradiation plan was evaluated in an interdisciplinary consensus meeting between the stereotactic neurosurgeon, a radiation oncologist experienced in SRS and the medical physicist. For radiosurgery the patient was immobilized on the Cyberknife^®^ treatment table (Accuray, Sunnyvale, California) by means of a custom-made aquaplast mask. 

### 2.3. Tumor Imaging and Volumetric Analysis

For volumetric analysis, we used axial T1-weighted, gadolinium-enhanced MRIs. Patients with follow-up MR images with slice thicknesses exceeding 3 mm were excluded from the study to minimize inaccuracy. On each image slice, the tumor cross-sectional area was calculated by delineation of the contrast-enhancing lesion via Osirix software (vs. 6, pixmeo). Tumor volume was subsequently calculated as the sum of the cross-sectional areas multiplied by the slice thickness, as previously described [14]. Tumor volume on the T1-weighted, gadolinium-enhanced MRIs performed for the SRS planning was defined as baseline and served in each case as 100%. Each follow-up MRI until the last available scan was used for the volumetric analysis. At each follow-up time point, the percentage volume change (%ΔV) from the baseline was also calculated.

For the purpose of the present analysis, we applied the RANO volumetric response criteria for meningioma [15] to the vestibular schwannomas (Table 1). Progressive disease (PD) was assumed if %ΔV exceeded 40% with additional continuous growth beyond 48 months of observation according to previous research [15,16,17]. Stable disease (SD) was stated if the tumor volume decreased by less than 65% ΔV or increased by less than 20% ΔV. A decrease of more than 65% ΔV was classified as partial response (PR). Complete response (CR) meant a complete remission of the entire tumor.

Additionally, pseudoprogression (PP) was defined as a tumor volume increase of more than 20% ΔV, followed by a decrease, finally resulting in SD or PR. We differentiated between early (volume peak within the first 12 months after SRS) and late PP (volume peak more than 12 months after SRS) (Table 1). Furthermore, morphological changes in the tumor in terms of loss of central contrast enhancement were documented (Table 2).

### 2.4. Statistical Analysis

Statistical analyses were performed using PRISM Ver. 8 (GraphPad software) and SPSS Vers. 25 (IBM). Descriptive statistics were used to characterize the patient population and treatment variables. Statistical significance was assumed if *p* < 0.05. The paired *t*-test was used to analyze volume changes over the observation period. A LogRank test or chi-square test was used to correlate onset of PP with new onset or deterioration of symptoms after SRS. In order to model the time course %ΔV(t) of the different types of volume changes in vs. after SRS, the following exponential equation was fitted to the data: %ΔV(t) = exp(–(A × t + B × t2)). 

## 3. Results

### 3.1. Patients and Treatments

We identified 106 patients who underwent Cyberknife^®^ SRS for a progressive vestibular schwannoma between 2013 and 2016. Thirty-two patients were excluded due to a follow-up comprising less than two MR images. Eleven patients were excluded due to MR images with a slice thickness of more than 3 mm. Finally, we were able to include 63 patients with a median tumor volume of 1.5 cm^3^ (range 0.1–8.6 cm^3^, see Table 2). The median clinical and radiological follow-up period was 66 months (range 24–104 months). Three patients had subtotal surgical resection prior to radiosurgery. 

### 3.2. Clinical Outcome

Clinical data regarding symptoms prior to and after treatment were available for all 63 patients (Figure 1). Twenty patients suffered worsening of one or more symptoms after Cyberknife^®^ SRS. In contrast, in thirteen patients symptoms improved. We did not find a correlation between the development of PP and aggravation of existing, or occurrence of new, symptoms after SRS (chi-square test, *p* = 0.339). Similarly, this was also true for new onset or deterioration of vertigo and balance disorders (LogRank, *p* = 0.636). Since only two patients in the collective suffered a loss of functional hearing after SRS, a correlation was statistically not feasible. 

### 3.3. Volumetric and Tumor Characteristics

Overall, 391 MR scans were volumetrically analyzed. The scans had a median slice thickness of 2.1 mm (range 1–3 mm). The mean tumor volume showed a significant decrease at all time points beyond 12 months after SRS (Figure 2). According to the predefined tumor response criteria regarding percentage volume change (%ΔV), four types of response were observed: partial response (PR, Figure 3A) in 36% (n = 23) of cases; stable disease (SD, Figure 3B) in 35% (n = 22) and pseudoprogression (PP) in 29% (n = 18) of cases. The latter was further divided into early PP (16%, n = 10, Figure 3C) and late PP (13%, n = 8, Figure 3D). The median time to onset of early PP was six months (range: 4-10) after SRS and of late PP, 15.5 months (range: 4–35). The overall median time to peak of tumor volume enlargement after SRS was 18 months (range: 4-61) in the case of early PP and 36 months (range: 20-61) in the case of late PP. The median ΔV% at the peak of transient enlargement was 57% (range: 20–225%). Eight out of ten patients in the early PP group and all patients in the late PP group had a transient enlargement exceeding 40% ΔV. The median duration of tumor enlargement (time to complete resolution) until regression was 12.5 months (range: 5–82). The mean time to resolve from peak was 8.9 months ± 4.8 (range 5–20) in the case of early PP, and 19.7 months ± 10.4 (range: 6–33) in the case of late PP. The difference between both groups was statistically significant (*p* = 0.011). The tumor enlargement resolved completely after 60 months in 94% of patients. An illustrative case with late PP is shown in Figure 4.

In 65% of the patients (n = 41), morphological changes with loss of contrast enhancement in the internal structure of the tumor were observed (Table 2). However, a correlation with early or late PP was not observed in the chi-square test (*p* = 0.378).

Possible influencing factors such as patient characteristics or radiation parameters are compared in Table 3. There were no statistical differences between the cohort of patients with PP or the cohort with PR and SD. 

## 4. Discussion

Pseudoprogression is a well-known phenomenon after SRS of VS. Nevertheless, the definition and reported duration or frequency of this phenomenon vary in the literature. To shed some light on this topic, we conducted an extensive literature search that included studies that met the following criteria: (1) single fraction SRS; (2) median follow-up of at least 24 months; and (3) volumetric response analysis. Eleven series with a total of 982 patients (Table 4) matched these criteria. While most of the available series reported results from the Gammaknife (GK), our series is one of the first retrospective surveys with detailed analyses of volumetric responses in vs. after robotic-guided SRS using the Cyberknife^®^. 

The incidence of pseudoprogression in our study was within the range of the other current studies (Table 4). Accordingly, pseudoprogression (PP) varied with an incidence of 4.7% [25] to 77% [19,20] and five-year tumor control varied between 87% [26] and 100% [27]. Some authors explain this disparity as being due to the large variation in observation periods in these studies [17]. Apart from this factor, different types of tumor volume measurements (2D or 3D) and different MRI protocols may hinder the general comparability of the results.

Comparable to our results, most patients seemed to develop a peak of volume increase between 6 months and 1 year after SRS (Table 4). However, in our series, seven patients had no early volume increase but developed a late transient tumor swelling, peaking at 3 or 4 years and then resolving very slowly. Other authors [16,17,23] also observed this late peak around 36 months after SRS (Table 4). Of note, this is the crucial time point for some authors to define loss of tumor control. For instance, Delsanti et al. [28] defined treatment failure as “a continuous growth for more than 3 years after radiosurgery”. Mindermann et al. [10] were even stricter and defined tumor progression in the case of a tumor volume increase of more than 20% after 24 months. 

Taking into account the present data, these definitions should be regarded with care, since all tumors still regressed spontaneously, even after 36 months or later. The case illustrated in Figure 4 demonstrates this observation and is in line with the results of Breshears et al. [16]. In about 90% of cases with transient enlargement, they observe a transient tumor enlargement at 3.2 years after SRS. Furthermore, both Breshears et al. [16] and Fouard et al. [17] reported that the transient enlargement resolved completely in 90% of the patients after up to 6.9 years, which is quite similar to our results, where a remission was observed in 94% of the patients after five years. 

### 4.1. Morphological Changes after SRS of Vestibular Schwannoma

Besides volume changes, we also observed morphological changes such as transient loss of central contrast enhancement in 65% of the patients, which is within the range of 45 to 83% reported in the literature [2,6,17,21,23,28]. The tumors showed loss of contrast only within the first year after SRS, but this phenomenon was not related to early or late PP, since in the patients with late PP, loss of contrast was also only apparent within the first year. These findings are similar to those of several other authors [17,21]. For instance, in the study of Fouard et al. [17], loss of contrast enhancement was also found in progressive tumors. This observation suggests that these morphologic changes may only represent an early radiation effect and are not necessarily predictive for early or late PP or PD.

So far, hardly any data are available about the mechanisms underlying morphological changes and PP. Iwai et al. [29] investigated the pathological changes in a group of patients who had salvage surgery after a median interval to radiosurgery of 28 months. Half of the patients had histologic findings of intratumoral hemorrhage, presence of macrophages, myxoid degeneration or necrosis; all changes were regarded as radiation-induced. Several authors have assumed that tumor enlargement after SRS is also due to radiation-induced tumor necrosis or chronic intratumoral hemorrhage [29,30]. Accordingly, here we suggest that during follow-up, the natural time course of tumor regression should be taken into account and surgical approaches should be limited to subtotal removal for functional preservation [29].

### 4.2. Risk Factors and Complications of Pseudoprogression 

Similar to the morphological changes after SRS, the pathological explanation for PP is not really known yet. However, vs. cells react with a combination of acute inflammation and vascular occlusion to radiation [29,31]. Therefore, it is, on the one hand, not surprising that morphological changes and also temporary enlargement occur. On the other hand, it is all the more incomprehensible that this phenomenon only affects a subset of treated VS. To date, possible risk factors for the development of PP are largely unknown or discussed controversially, although prognostic assessment for PP occurrence would be important for post-interventional management. Regarding patient-, tumor- or treatment-related factors such as age, tumor volume and radiation dose, we did not find any influencing factors. This goes in line with the majority of the available studies [6,18,21,23,32], with the exception of the study of Kim et al. [24], which showed that solid-type tumors had a higher probability of developing PP. This might be due to the fact that cystic tumors have less tissue that can respond to radiation.

New insights can probably be gained from MRI-based radiomics. A study by Langenhuizen et al. [33] detected PP in 38 out of 99 patients. A correlation between patient- and treatment-related factors and PP was not evident. However, textural features from MRI scans derived from the three-dimensional gray-level co-occurrence matrices (GLCM) showed a prognostic value for PP with a sensitivity of 0.82 and specificity of 0.69. These findings suggest that MRI-based tumor texture analysis provides information that could be used to predict PP and serve as a basis for individualized vs. treatment and FU strategy, especially in patients with large VS in which the phenomenon of PP is most relevant.

In our series, we did not find a significant association between postradiosurgical clinical deterioration (cranial neuropathies, ataxia and hydrocephalus) and PP. This is in line with a majority of studies [6,10,11,16,17,32]. Nevertheless, a relationship between onset of new cranial neuropathies or hydrocephalus was present in some studies within the first year after SRS [19,34,35]. For example, Pollock et al. [35] reported adverse effects in 20% of patients with tumor growth. However, the number of patients with AEs without tumor growth was not shown in that study. Nagano et al. [19] saw an association with tumor volume increase of more than 30% associated with a significantly higher rate of cranial nerve impairment (e.g., facial hemispasm, facial weakness and facial dysesthesia). Aoyama et al. [34] reported a rate of post-SRS hydrocephalus, with 11% of patients and a median of 11.3 months after the treatment [34]. Tumor expansion and tumor size measuring 30 mm or greater are considered as risk factor for the new onset of hydrocephalus. In general, it is not so much the size but rather a higher protein concentration of cerebrospinal fluid after SRS [36] which leads to a blockage of arachnoid granulations and is therefore seen as the main risk factor. This hypothesis also explains better why smaller tumors and cases without PP can also develop hydrocephalus. However, in many other series [2,4,6,7,11,16,17,19,20,21], the development of hydrocephalus plays no or only a minor role.

### 4.3. Development of RANO Criteria for vs. after SRS

The phenomenon of pseudoprogression seems to be a quite typical phenomenon for vestibular schwannoma and is almost absent in other benign tumors such as meningiomas, hemangiopericytomas, pituitary adenomas or glomus tumors. However, in meningiomas, PP after SRS occurs in 5–11% and is typically associated with morphological changes indicating intratumoral necrosis [37]. In VS, the situation is probably more complex, as in many tumors, PP occurs without morphological changes. 

Regarding benign brain tumors, RANO criteria are, so far, available only for meningioma [15], but not for VS. Thus, apart from arbitrarily defined criteria for distinguishing progression from pseudoprogression in VS, clear recommendations for response assessment would be helpful (Tab 3.). Even the well-established guidelines for the treatment of treatment of VS [1,5] do not provide a consistent and clear recommendation, only the guideline of the International Society for Radiosurgery (ISRS, [38]) mentions that PP should be considered within the first three years after SRS. 

In order to fill this gap, we here propose additional response criteria to existing RANO criteria [15] to overcome the ambiguities and different results of the various studies. These criteria use the well-established RANO terms of “stable disease” (SD), “partial response” (PR) and “progressive disease” (PD), as these allow a clearer understanding of the tumor response after treatment. The percentage volume changes of the respective category are taken from the RANO criteria for meningioma [15]. In addition, the time interval in which volume changes occur should be considered. Therefore, we suggest the new category, “pseudoprogression” (PP), divided into early and late occurrence. 

Some studies on PP in vs. suggest >20% for the definition of PP. We follow this definition. On the other hand, the RANO criteria for meningiomas define PD as >25% volume increase [15]. Since 20% and 25% are very close to each other and late PP can still be present after 36 months, it is difficult to separate both definitions. Especially, the temporal overlap makes it difficult to define a valid threshold. In our study, the best discriminatory power is 40%, because at >48 months all but three patients with late PP fell below this threshold. An increase of the threshold value, on the other hand, bears the risk of overlooking true progression. Furthermore, we follow the results of the study by Fouard et al. [17]. They found that all patients with true progression had a volume increase of 37-43% at 36-48 months, so a value of 40% seems plausible. Further studies with a larger cohort might show that the current thresholds should be set differently after all. 

Furthermore, our proposed RANO criteria take the duration of tumor enlargement into account, and as such, are in line with our results and results from the existing literature (Table 4).

Nevertheless, any response assessment criteria have the problem that they must tell stable from progressive disease also from the clinical viewpoint. As there were only mild clinical deteriorations associated with pseudoprogression in our cohort, salvage treatment was never necessary. If a clinical deterioration correlates with the imaging changes and appears to be uncontrollable, a new intervention may be required. However, surgery as a salvage treatment should always be considered carefully to avoid treating too early and to allow the tumors to regress spontaneously. 

### 4.4. Conclusion and Practical Implications of Proposed RANO Criteria for VS

The criteria proposed here are primarily intended to provide guidance for post-interventional management. Patients with vestibular schwannomas scheduled for radiosurgery should be informed about the phenomenon of early and late pseudoprogression. High-resolution contrast-enhanced T1-weighted MRI should be used as gold-standard for follow-up imaging, as this sequence allows the best visualization of the tumor. Annually, MRI exams are recommended during the first 5 years and can be reduced to an interval of 2–3 years if the tumor volume is stable or smaller than before treatment [17]. A slice thickness of 2 mm or less should be used because volume measurement errors increase exponentially with the slice thickness [39]. In clinical routine, the measurement of transverse tumor diameter may be adequate in most of the cases [40]. Any case with suspected progressive disease in MRI controls should be examined in short intervals (e.g., 3–6 months) by thin-slice MRI and 3D volumetry of the tumor, since the latter can reduce measurement errors [41,42,43]. Even in large tumors with or without onset of new symptoms, caution regarding salvage surgery or radiotherapy is required.

## 5. Conclusions

In light of the newly developed response criteria and the inconsistency of the actual recommendations, the criteria for tumor progression often proposed in the literature (e.g., tumor volume increase of more than 10–20%, tumor growth after 3 years, no return to pretreatment volume after transient swelling) are, in our opinion, no longer valid and should not be used. Additionally, longer observation periods after SRS are strongly recommended based on these findings.

## Figures and Tables

**Figure 1 cancers-15-01496-f001:**
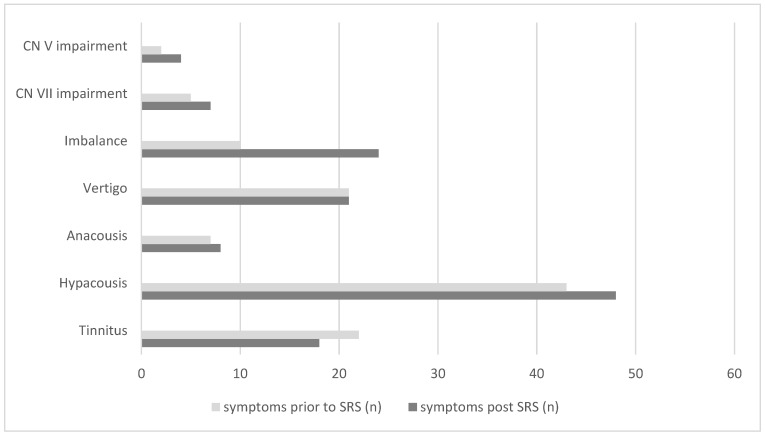
Symptoms prior to and post Cyberknife® SRS in a collective of 63 patients with progressive VS.

**Figure 2 cancers-15-01496-f002:**
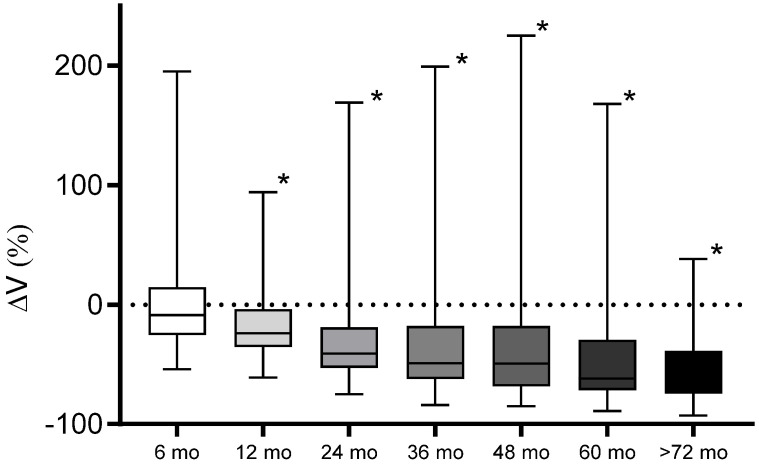
Mean tumor volume during the follow-up period. Twelve months after SRS, the mean tumor volume of the collective was significantly decreased (marked with *) compared to the baseline (6 months: *p* = 0.91, 12 months: *p* < 0.0001, 24 months: *p* < 0.0001, 36 months: *p* = 0.018, 48 months: *p* = 0.021, 60 months: *p* < 0.0001, >72 months: *p* < 0.0001).

**Figure 3 cancers-15-01496-f003:**
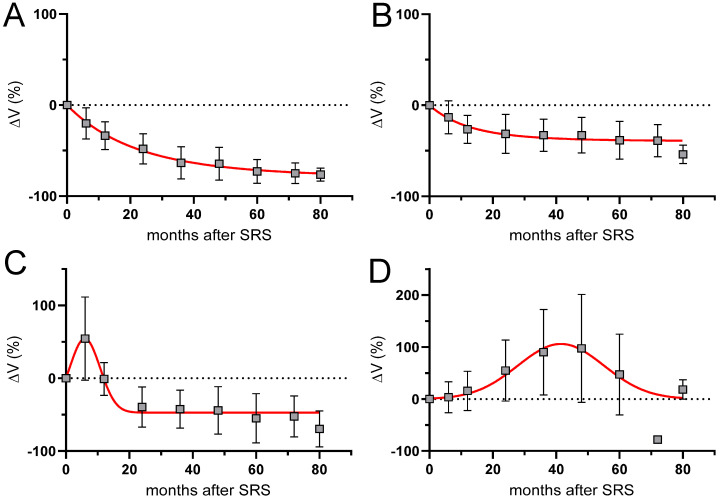
Comparison of mean %ΔV during follow-up after SRS of VS. (**A**): Partial response; (**B**): Stable disease; (**C**): Early pseudoprogression; (**D**): Late pseudoprogression.

**Figure 4 cancers-15-01496-f004:**
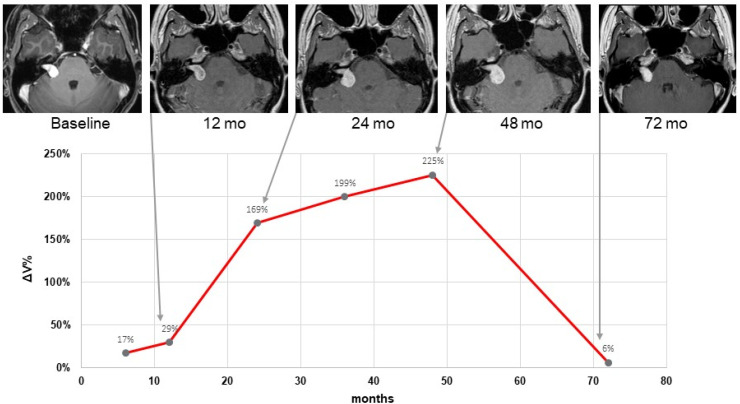
Illustrative case of a 56-year-old woman with a vestibular schwannoma that showed an increase in volume and was classified as late pseudoprogression. At the baseline, Cyberknife^®^ SRS with a radiation dose of 13 Gy was applied. During follow-up, a continuous volume increase could be observed with a maximum of %ΔV = 225 after 48 months. Afterwards, until 72 months, the tumor volume regressed almost completely to baseline.

**Table 1 cancers-15-01496-t001:** Proposed RANO criteria for assessing tumor response after SRS of VS.

Criterion	CR Complete Response	PR Partial Response	SD Stable Disease	PP Pseudoprogression	PD Progressive Disease
(1) Target lesion	None	≥65% decrease in volume relative to baseline	<65% decrease relative to baseline but <20% increase in volume relative to nadir	>20% increase in volume relative to baseline followed by decrease within 24–36 months	≥40% increase in volume relative to baseline and/or change of Koos grade I or II to III or IV
(2) Onset of volume increase after SRS	n.a.	n.a.	n.a.	Early peak: <12 months Late peak: >12 months	>36–48 months
(3) Clinical status (besides hearing deterioration and vertigo)	Stable or improved	Stable or improved	Stable or improved	Stable or improved	Stable or deteriorated
Requirement for response	All or (1) alone	All or (1) alone	All	All	All

**Table 2 cancers-15-01496-t002:** Patient and treatment characteristics, clinical data and response criteria of the observed cohort after Cyberknife^®^ SRS. Unless mentioned otherwise, data are presented as mean with SD and range in brackets.

**Patient Characteristics**	
Total no. of patients	63
Gender (m:f)	25:38
No. of Koos-Grade	KoosI: 6 KoosII: 45 KoosIII: 11 KoosIV: 1
Age (years)	56 ± 14 (range: 20–82)
Tumor volume (cm^3^)	1.5 ± 1.4 (range: 0.1–8.6)
Median radiological and clinical FU (months)	66 (range: 24–103)
**Radiation Parameters**	
Marginal dose (Gy)	13.0 ± 0.2 (range: 12–13)
Dose prescription, isodose (%)	80 ± 3.5 (range: 65–81)
Coverage (%)	99.6 ± 0.9 (range: 94.7–100)
Dmax	16.25 ± 0.7 (range: 15–18.6)
Dmean	14.89 ± 0.3 (range: 14–15.8)
Dmin	12.94 ± 0.4 (range: 11–13.2)
nCI	1.19 ± 0.07 (range: 1.08–1.44)
**Response/Pseudoresponse Criteria**	
Loss of central contrast enhancement	41 (65%)
CR—complete response	0
ePP—early pseudoprogression	10 (16%)
lPP—late pseudoprogression	8 (13%)
SD—stable disease	22 (35%)
PR—partial response	23 (36%)
PD—progressive disease	0

**Table 3 cancers-15-01496-t003:** The comparison of pseudoprogression (PP) and partial response/stable disease (PR+SD) of the observed cohort after Cyberknife® SRS showed no significant difference (I > 0.05) with regard to patient and treatment characteristics.

Patient Characteristics	PR + SD	PP	*p*-Value
No. of patients	**45**	**18**	
Age (years)	**57** ± 14.3	**54** ± 13.2	*p =* 0.731
Tumor volume (cm^3^)	**1.62** ± 1.5	**1.15** ± 0.7	*p* = 0.057
follow-up (months)	**61.5** ± 21.6	**64.3** ± 21.1	*p* = 0.083
**Radiation Parameters**			
Marginal dose (Gy)	**13**	**13**	
Dose prescription, isodose (%)	**78.02** ± 3.7	**78.9** ± 2.7	*p* = 0.082
Coverage (%)	**99.36** ± 1.1	**98.8** ± 1.4	*p* = 0.087
Dmax	**16.6** ± 0.7	**16.5** ± 0.6	*p* = 0.223
Dmean	**14.9** ± 0.2	**14.8** ± 0.3	*p* = 0.111
Dmin	**12.8** ± 0.4	**12.8** ± 0.2	*p* = 0.158
nCI	**1.17** ± 0.63	**1.22** ± 0.09	*p* = 0.051

**Table 4 cancers-15-01496-t004:** Summary of pseudoprogression after single fraction SRS of vs. evaluated by volumetric analysis in former studies. Incidence and time course of pseudoprogression.

Study	n	Mean FU (months)	Radiation Technique	Tumor Volume (ml)	Incidence of PP (%)	Definition of PP	Mean Volume Increase (%)	Median Yime to Peak for PP (months)	Late Peak (% of Collective @ Median Ttime to Peak)	Duration of PP
Yu, 2000 [18]	91	22	GK		63	NA	20	6	NA	
Nakamura, 2000 [6]	78	34	GK	0.6	41	Volume increase of more than 20%	NA	12	6.4% @ 24–36 months	24 mo
Nagano, 2008, 2010 [19,20]	87	65	GK	2.5	77	Volume increase > 10%	58	8.6	NA	90% resolved after 5 yr
Van de Langenberg, 2011 [21]	17	40	LINAC	2.09	54	Volume increase >19.7%	NA	5 (3-17)	NA	15 (8–27) mo
Hayhurst, 2012 [11]	75	29	GK	1.7	23	Volume increase > 10%	23	NA	NA	NA
Kim, 2013 [22]	60	42	GK	0.34	47	Volume increase > 10% within a year	NA	NA	NA	NA
Mindermann, 2014 [10]	235	62	GK	1.85	NA	Volume increase > 20% within 24 months	NA	6-18	NA	12-18
Matsuo, 2015 [23]	44	165	LINAC	2.38	54.5	Volume increase > 20% within 24 months	88	9	7.1% @ 39.6months	
Kim, 2017 [24]	235	34	GK	2.2	18	Volume increase > 20% within a year	NA	7 (4–55)	NA	NA
Breshears, 2019 [16]	18	49	GK	0.74	42	Volume increase at any time during FU followed by reduction	49	12.5	10% @36–48 months	28.8 mo, 90% resolved at 6.9 yr
Fouard, 2021 [17]	42	83	GK	0.69	63.5	Volume increase > 13% at any time during FU followed by reduction	64	6-12 mo	17% @ 36–48 months	24 mo,
Our study	63	66	CK	1.5	29	Volume increase > 20% at any time during FU followed by reduction	57	18	12.7% @ 36 months	12.5 mo, 90% resolved after 4 yr

## Data Availability

Not applicable.

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
