# Peer review of "Pseudoprogression of Vestibular Schwannoma after Stereotactic Radiosurgery with Cyberknife®: Proposal for New Response Criteria"

_cancers, 2023, doi:10.3390/cancers15051496_

Round 1

Reviewer 1 Report

The idea of ​​the study is appreciable, the results are well presented and the proposal to include schwannomas in the RANO criteria is positive. I would suggest diversifying and comparing the results (case history is quite limited) both with larger populations treated with Gamma knife and with other forms of Linac, to evaluate the incidence and trend of the pseudoprogression on different machines, perhaps through a multicenter study . This would allow for greater accuracy in the RANO criteria. I would elaborate on the paragraph of the discussion concerning the causes that are at the origin of the pseudoprogression, since this is the focus of the work. I would also relate pseudoprogression to the risk of hypersecretic hydrocephalus or other complications. I would also deepen the relationship between pseudoprogression, growth even in the case of more aggressive lesions, such as NF2-related neuromas.

Author Response

We thank the reviewer for the thoroughly review of our manuscript. However, we add explanations in the manuscript.

The idea of ​​the study is appreciable, the results are well presented and the proposal to include schwannomas in the RANO criteria is positive. I would suggest diversifying and comparing the results (case history is quite limited) both with larger populations treated with Gamma knife and with other forms of Linac, to evaluate the incidence and trend of the pseudoprogression on different machines, perhaps through a multicenter study. This would allow for greater accuracy in the RANO criteria.

We thank the reviewer for this idea for further research in the field. A multicenter study in this regard will be planned in the near future.

I would elaborate on the paragraph of the discussion concerning the causes that are at the origin of the pseudoprogression, since this is the focus of the work. I would also relate pseudoprogression to the risk of hypersecretic hydrocephalus or other complications.

For this purpose, the chapter “Risk factors and complications of pseudoprogression” was revised accordingly and the complications and causes of PP were addressed. The changes can be found in the discussion section in lines 50-55 and 73-90 on page 2/19.

I would also deepen the relationship between pseudoprogression, growth even in the case of more aggressive lesions, such as NF2-related neuromas.

NF-2-related VSs are histologically different from sporadic tumors and usually have a poorer outcome after SRS (Sharma et al 2010 J Neurooncol). Thus, the results in patients with sporadic VSs cannot be generalized to patients with NF-2, and vice versa. That´s why did not include patients in this study and our modified RANO criteria account only for sporadic VS.

Reviewer 2 Report

Good analysis on a known phenomenon.

Questions: 1) Did any patient with PP had a tumor volume increase >40%? 2) How did you come up with 40% as a criterion for PD? 3) 94% of patients with PP had resolution of their tumor enlargement. What happened to the remaining 6%? Did they ultimately had PD?

Author Response

We thank the reviewer for the thoroughly review of our manuscript. However, we answer the reviewers questions in the following and add explanations in the manuscript.

Questions:

1) Did any patient with PP had a tumor volume increase >40%?

Yes, eight out of ten patients in the early PP group and all patients in the late PP group. We add this values with one sentence in the results section “Volumetric and tumor characteristics”.

2) How did you come up with 40% as a criterion for PD?

The major problem of late pseudoprogression is the adjacency to PD.  Some studies on pseudoprogression (PP) in VS suggest >20% for the definition of PP.  We agreed with this definition. On the other hand, the RANO criteria for meningiomas define PD as >25% volume increase. Since 20% and 25% are very close to each other and late PP can still be present after 36 months, it was difficult to separate both definitions. Especially the temporal overlap made it difficult to define a valid threshold. In our study, the best discriminatory power was 40%, because at >48 months all but three patients with late pseudoprogression had fallen below this threshold. An increase of the threshold value, on the other hand, bears the risk of overlooking true progression. Furthermore, we followed the results of the study by Fouard et al. They found that all patients with true progression had a volume increase of 37-43% at 36-48 months, so a value of 40% seems plausible. A larger cohort might show that the threshold should be set differently after all.

However, not only the threshold value itself is relevant, but also the time of occurrence and the time course of the volume changes. We add the explanation of this threshold in the discussion in line 115  up to 125 on page 3/19.

3) 94% of patients with PP had resolution of their tumor enlargement. What happened to the remaining 6%? Did they ultimately had PD?

94% of patients had a resolution of tumor enlargement after 60 months. The remaining four patients showed further volume decrease after 72 months follow-up, so that finally at last follow-up none of the patients showed increase of tumor volume or a tumor volume >40%.
